# Count-guided Weakly Supervised Localization Based on Density Map

## Abstract

Weakly supervised localization (WSL) aims at training a model to find the positions of objects by providing it with only abstract labels. For most of the existing WSL methods, the labels are the class of the main object in an image. In this paper, we generalize WSL to counting machines that apply convolutional neural networks (CNN) and density maps for counting. We show that given only ground-truth count numbers, the density map as a hidden layer can be trained for localizing objects and detecting features. Convolution and pooling are the two major building blocks of CNNs. This paper discusses their impacts on an end-to-end WSL network. The learned features in a density map present in the form of dots. In order to make these features interpretable for human beings, this paper proposes a Gini impurity penalty to regularize the density map. Furthermore, it will be shown that this regularization is similar to the variational term of the $\beta$-variational autoencoder. The details of this algorithm are demonstrated through a simple bubble counting task. Finally, the proposed methods are applied to the widely used crowd counting dataset the Mall to learn discriminative features of human figures.

## 1 Introduction

Deep convolutional neural networks (CNN) have significantly pushed the frontier of image processing. Trained with enormous amount of data, CNNs have surpassed human performance in many object detecting tasks. However, humans are still ahead in many aspects. One of them is the segmentation based on abstract knowledge: once a human can tell what is in an image, he/she can easily crop the objects out. Researches have been exploring similar potentials of CNNs. Related techniques are referred to as Weakly Supervised Localization (WSL), in which models are trained with only image-level labels but expected to give the positions of objects at once. Most of existing WSL methods are guided by class labels and realized in two different ways.

The first is the Multiple Instance Mining (MIM). MIM is based on multiple instance learning (MIL), which trains a model to identify multiple instances from incomplete labels (Babenko, 2008; Siva et al., 2012). For object detection, the regions around the target have the highest influence on the label decision. By evaluating the contributions of different regions, the location of the target can be determined. A straightforward approach to find the relevant regions is to compute the derivative of the decision with respect to the input image (Simonyan et al., 2013). But the found regions can also be the ones that introduce the most uncertainty to the result. A more effective approach is to use an attention mechanism, in which one detector proposes potential regions according to a classifier's evaluation of contribution (Gokberk Cinbis et al., 2014; Song et al., 2014a; Bilen et al., 2014; Ren et al., 2016; Cinbis et al., 2017).

The second way to WSL is to train the locations as latent variables. Zhou et al. reported that a deep CNN can maintain spatial information through layers (Zhou et al., 2016). Based on this observation they proposed the class activation map (CAM) which averages every channel of a CNN's final layer and feeds the averages to a soft-max classifier. This structure is analogous to the density map used for object counting. The density map method annotates each object by one pixel with a value of one and computes the count by summing up the map (Lempitsky & Zisserman, 2010).

Inspired by the similarities between the CAM and the density map, we generalize WSL to counting models based on density map. Our method trains the density map as latent variables of an end-

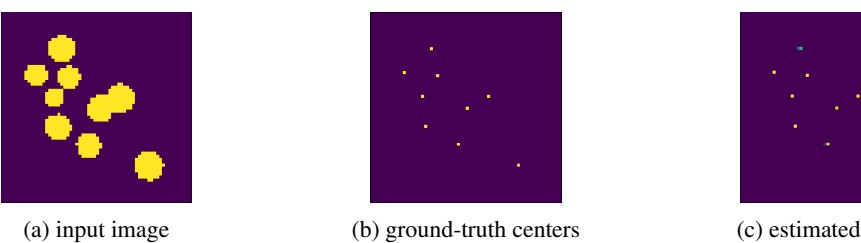

(a) input image          (b) ground-truth centers          (c) estimated centers

Figure 1: Bubble image and supervised localization

to-end CNN by giving only the final count of objects. It is observed that the trained density map can roughly capture the positions of objects, but the accuracy is degraded by the convolution and pooling operations. This paper will discuss the impacts of these two operations on an end-to-end WSL network and propose techniques to improve accuracy. The density map counts objects by detecting discriminative features. To improve interpretability of the found features, we introduce a Gini impurity penalty to regularize the density map. Furthermore, the similarities and differences between this regularization and the variational term of the $\beta$-variational autoencoder (VAE) (Higgins et al., 2017) will be discussed. Throughout this paper, we will use a simple simulated bubble dataset to demonstrate basic ideas and a widely used crowd-counting dataset the Mall Loy et al. (2013); Chen et al. (2013); Change Loy et al. (2013); Chen et al. (2012) to show the methods' effectiveness in practice.

## 2 RELATED WORK

### 2.1 MULTIPLE INSTANCE MINING

Coarsely labelled images are labelled only by the class of the main object. However, due to the usage of softmax function as output and the cross entropy for training, a classifier has the potential to identify multiple instances (Babenko, 2008). A MIM WSL method trains a detector to propose either a bounding box (Bilen & Vedaldi, 2016; Tang et al., 2017) or a segmentation (Kolesnikov & Lampert, 2016; Wei et al., 2017) that highlights the object in the image. A few candidates for bounding boxes or segmentations are initialized at the beginning, and the detector is trained by trial and error. This always results in a non-convex optimization problem that easily converges to local optimum. Solutions for this problem focus on both the initialization (Deselaers et al., 2012; Siva et al., 2013; Shi et al., 2013; Song et al., 2014b) and the training strategy (Crandall & Huttenlocher, 2006; Cinbis et al., 2017; Li et al., 2016). This attention mechanism can also be realized in an adversarial style (Wei et al., 2017; Shen et al., 2018; Bartz et al., 2018).

Among all MIM methods, the min-entropy latent model (MELM) and the count-guided weakly supervised localization (C-WSL) are mostly related to this work. The MELM (Wan et al., 2018) uses the min-entropy metric to measure and reduce the randomness of localization proposals. Likewise, our method leverages the Gini impurity to decrease the complexity of the latent density map. The C-WSL (Gao et al., 2018) takes advantage of per-class object count labels to reduce label cost and improve performance. Even though it is also guided by count, it applies a MIM paradigm in contrast to our end-to-end density-map-based approach.

### 2.2 DENSITY MAP COUNTING

Density map is a counting technique based on detection (Lempitsky & Zisserman, 2010). Combined with CNNs, it has achieved state-of-the-art performance in tasks like cell (Xie et al., 2018), crowd (Zhang et al., 2016), and wild animal (Arteta et al., 2016) counting. Commonly, counting CNNs have the structure of a Fully Convolutional Neural Network (FCNN) (Xie et al., 2018), because images for counting usually contain similar objects spreading all over the frame, and a FCNN can process images of arbitrary sizes.

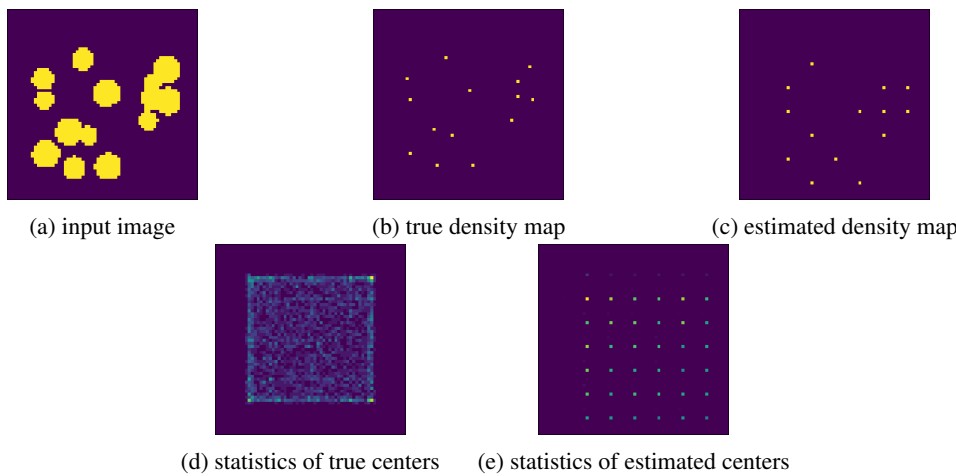

(a) input image        (b) true density map        (c) estimated density map

(d) statistics of true centers      (e) statistics of estimated centers

Figure 2: Weakly supervised Localization with max-pooling

### 2.3 CLASS ACTIVATION MAPPING

The CAM localizes the positions of discriminative features by applying the global average pooling (GAP) to the final layer of a CNN. Because it is trained in a MIL way, the CAM can identify multiple features in a image (Zhou et al., 2016). A trained CAM can also be used for proposing candidates in a MIM method (Chen et al., 2018). To some extent, CAM is analogous to density map counting, as averaging the channels is equivalent to integrating the density map. On the other hand, CAM turns a CNN with fully connected layers to a FCNN.

### 2.4 VARIATIONAL AUTOENCODER

VAE (Kingma & Welling, 2013) or more generally $\beta$-VAE (Higgins et al., 2017) is an unsupervised data exploring method. The hidden features of the dataset are learned by training a network to compress and reconstruct its input. Experiments and theories have shown that the variational term of the VAE forces an auto-encoder to learn more disentangled latent features (Higgins et al., 2017; Alemi et al., 2016; Pu et al., 2016; Kim & Mnih, 2018; Burgess et al., 2018). In the following work, we will use a similar technique to make the density map yield interpretable results.

## 3 COUNT-GUIDED WSL BASED ON DENSITY MAP

In this section, we investigate the count-guided weakly supervised localization based on density map. We first describe the basic counting structure that takes the density map as latent variables and then discuss the impact of pooling and convolution on this end-to-end WSL network. Finally, we propose the Gini impurity penalty to control the representation of a density map and show its relation with the $\beta$-VAE.

### 3.1 BASIC COUNTING STRUCTURE

The feature extractor is a FCNN that outputs a map $f_\Theta(I)$ of the same size as the input image $I$, where $\Theta$ are the parameters of the network. The map is activated by the rectified linear unit (ReLU) and then goes through a hyperbolic tangent function to make sure all its elements are either zero or less than one; i.e. $D = tanh(f)$. Finally, the estimated count $\hat{c}$ is the integral of the density map:

$$\hat{c} = \sum D_\Theta(I) \tag{1}$$

At the training stage, we only provide the final count $c$ of objects as ground truth.

$$\hat{\Theta} = \arg \min_\Theta \mathbb{E}_{data}[(c - \hat{c})^2] \tag{2}$$

Table 1: FCNN for bubble counting

| operations | kernel | size | channels |
|---|---|---|---|
| conv | 3×3 | 64×64 | 32 |
| conv | 3×3 | 64×64 | 64 |
| max-pooling | 3×3 | 32×32 | 64 |
| conv | 3×3 | 32×32 | 128 |
| max-pooling | 3×3 | 16×16 | 128 |
| conv | 3×3 | 16×16 | 512 |
| max-pooling | 3×3 | 8×8 | 512 |
| deconv | 3×3 | 16×16 | 128 |
| deconv | 3×3 | 32×32 | 64 |
| deconv | 3×3 | 64×64 | 32 |
| conv ($f(I)$) | 3×3 | 64×64 | 1 |

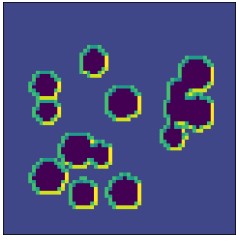

Figure 3: First layer output
(yellow for higher values)

The goal is to train the latent density map for localization:

$$\hat{\Theta} = \arg\max_{\Theta} \mathbb{E}_{q_{\Theta}(\mathrm{D}|\mathrm{I})}[\log(p(\mathrm{c}|\mathrm{D}))] \tag{3}$$

In terms of training, equation 2 and 3 are identical. Following the notation of VAE (Burgess et al., 2018), we expose the hidden density map in 3 for further analysis.

## 3.2 SPATIAL INFORMATION LOSS

Conventionally, a CNN applies pooling layers to reduce computations and extract abstract features. However, as a down-sampling operation, pooling does harm to the spatial information going through a deep network. Here we use a simple bubble counting task to study the impact of pooling on this end-to-end WSL network.

### 3.2.1 SETUP OF BUBBLE COUNTING

The simulated data is a set of 64×64 binary images with a randomly selected number of 1 to 15 bubbles in the frame, as shown in Figure 1a. The bubble centers are located within a square as shown in Figure 1b and Figure 2d, and their radii are 3 to 5 pixels. Slight overlaps are allowed. The goal is to localize the bubble centers by providing only the counts for training. Table 1 shows the structure of the FCNN for this test. It uses max-pooling following the second, third, and fourth convolutional layers to down-sample features. After reaching the bottleneck, it uses transposed convolutions (deconvolution) for staged up-sampling to finally match the input size at the output.

### 3.2.2 SUPERVISED LOCALIZATION

Firstly, we make sure this FCNNs capacity is sufficient for this localizing task. Both the ground-truth density maps D and the final counts c are provided for training:

$$\hat{\Theta} = \arg\min_{\Theta} \mathbb{E}_{data}[(\mathrm{c} - \hat{\mathrm{c}})^2 + \sum (\mathrm{D} - \hat{\mathrm{D}})^2] \tag{4}$$

We generate a mini-batch of 10 samples for every training step and a set of 500 examples for testing. After $3 \times 10^5$ evaluations, the average pixel-by-pixel error of the density map converges to a sufficiently small value of 0.001. An example of the estimated centers is shown in Figure 1c. This demonstrates that the proposed network can count as well as localize bubbles.

### 3.2.3 WEAKLY SUPERVISED LOCALIZATION

Now we use the same training strategy to train the network again but provide it only with the final counts (equation 2). After $3 \times 10^5$ epochs, the counting error converges to 0.02. But, as shown in Figure 2c, the localization is not as good as in the case of supervised localization and shows two artifacts:

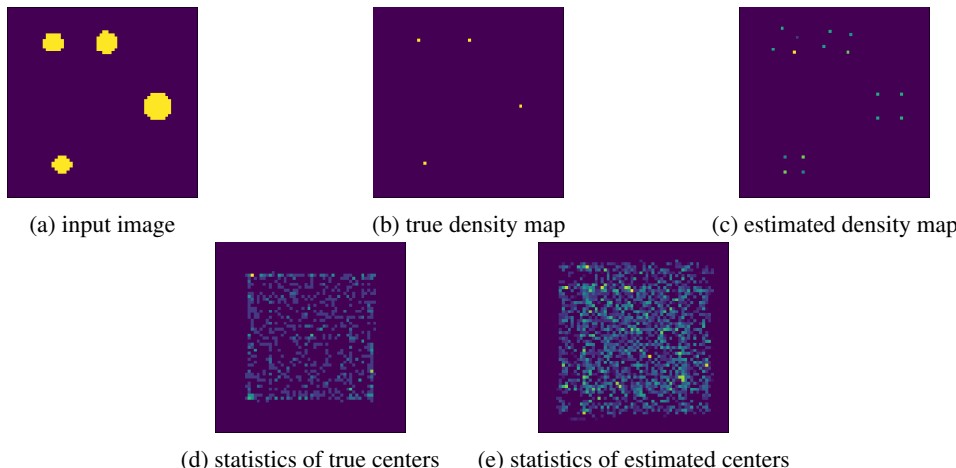

(a) input image         (b) true density map         (c) estimated density map

(d) statistics of true centers     (e) statistics of estimated centers

Figure 4: Weakly supervised Localization robust to rotation

Firstly, the centers are relocated to match a low resolution grid. Figure 2e compiles the statistics of 500 test samples to draw this grid clearly. This spatial information loss is caused by the pooling layers, and it cannot be recovered by up-samplings. In contrast, this loss of detail does not happen in the case of supervised localization (Figure 1), because the presence of the true centers in the labels forces the network to retain position information even through down-sampling processes. Due to this drawback, we will not use any pooling layers in the following WSL models.

Secondly, the estimated centers as a whole drift to the right bottom corner of the density map. We believe this is because CNNs rely on edge detection, and our training data is homogeneous. The output of the first convolutional layer (sum of all channels) in Figure 3 shows that the trained kernels highlight the right bottom edges more than other sides. Going through all the layers, this bias will finally show in the density map. Zhou et al. (2016) discussed that the GAP, unlike the max-pooling, encourages the network to identify the full extent of the object instead of certain edges. However, in this count-guided case, the bias happens before any pooling layers (Figure 3), and the averaging (sum) cannot recover the correct positions of the centers (Figure 2). Thus, we resort to a direct solution to tackle this problem.

### 3.3 ROTATION ROBUSTNESS

In order to concentrate the learned features at the center of an object, we encourage the network to be robust to rotations. To accomplish this, the input image is rotated by $90°$, $180°$, and $270°$, and the differences between rotations are to be reduced. Denote the rotated image as $I_{degree}$, its density map as $D_{degree}$, and its count as $c_{degree}$. The loss function comes to

$$\mathbb{E}_{data}[\sum_k (c - \hat{c}_k)^2 + \sum_{i,j} (\hat{D}_i - \hat{D}_j)^2] \quad i, j, k = 0°, 90°, 180°, 270° \quad (5)$$

We use the FCNN structure in Table 1 for feature extraction, but the down and up sampling operations are discarded for accurate localization. The results are shown in Figure 4. Compared with Figure 2, the density map is no longer limited by the lower resolution grid because of the absence of pooling layers. However, one bubble is represented by four dots because we turned the training images four times to make the network immune to rotations. To further centralize the dots, we propose a Gini impurity penalty in the following section.

### 3.4 GINI IMPURITY PENALTY

Intuitively, using more dots in a density map increases the maps impurity. One of the widely used metrics of impurity is the Gini impurity (Fayyad & Irani, 1992). It is defined as the probability of one state changing to its opposite. For a two-dimensional density map, it is

$$G(D) = \sum p(D)(1 - p(D)) = 1 - \sum p^2(D) \quad (6)$$

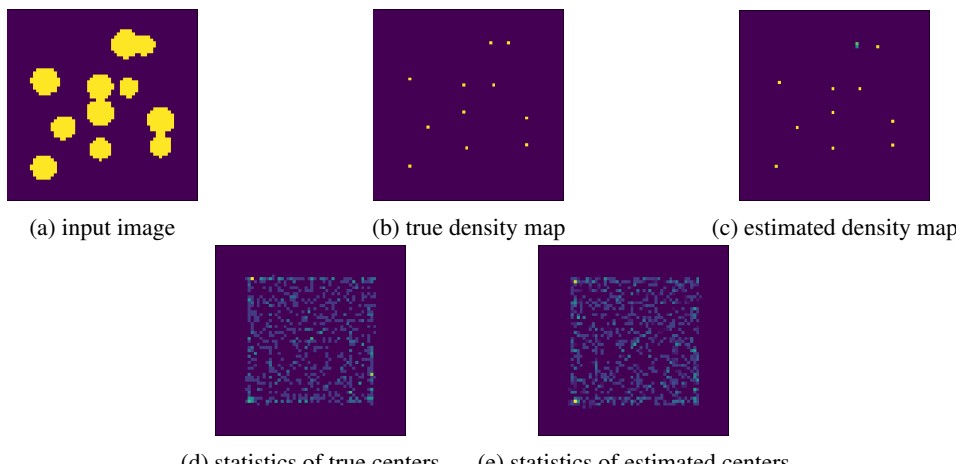

(a) input image       (b) true density map       (c) estimated density map

(d) statistics of true centers       (e) statistics of estimated centers

Figure 5: Weakly supervised localization with rotation robustness and Gini impurity penalty

where $p(\mathrm{D})$ is a 2-D distribution obtained by normalizing the density map:

$$p(\mathrm{D}) = \frac{D}{\sum D + \epsilon} \tag{7}$$

We add a small number $\epsilon$ to the denominator in case the integral is zero in practice.

Because all the elements of the density map are less than one, if two density maps have the same integral, the one with more non-zero elements will have higher Gini impurity. For example, in Figure 4, the sums of the true and estimated density map are very close (4.00 and 4.08), but their Gini impurities are 0.75 and 0.93, respectively. We can reduce the amount of non-zero pixels by decreasing the Gini impurity of a density map. The loss function becomes

$$\mathbb{E}_{data}\left[\sum_k (\mathrm{c} - \hat{c}_k)^2 + \sum_{i,j}(\hat{D}_i - \hat{D}_j)^2\right] + \beta \sum G(\hat{D}_k) \quad i,j,k = 0°, 90°, 180°, 270° \tag{8}$$

where $\beta$ is a positive real hyper-parameter that controls this regularization term.

Like before, the FCNN in Table 1 without pooling layers are used for feature extraction. $\beta = 10.0$ has been applied to regularize the density map. The outcomes are shown in Figure 5. Compared with previous results (Figure 2 and Figure 4), the detected centers are no longer limited by a shifted low-resolution grid, and the amount of non-zero dots matches the exact count to a great extent.

Equation 8 indicates that the Gini impurity penalty is related to the $\beta$-VAE. Here we study the case without the rotations. Given the count of objects, the Gini impurity of the true density map is a fixed number:

$$G(D) = 1 - c \cdot \frac{1}{c^2} = 1 - \frac{1}{c} \tag{9}$$

After training, we assume that $\sum \hat{D} = c$. Because all elements of the density map are between zero and one, there is

$$G(\hat{D}) \geq G(D) \tag{10}$$

Thus, minimizing the Gini impurity can be achieved by reducing the information gain

$$\Delta H(\hat{D}, D) = G(\hat{D}) - G(D) \tag{11}$$

which is equivalent to minimizing the Kullback-Leibler (KL) divergence between the prior distributions $p(\mathrm{D})$ and the posterior distribution $q_\Theta(\hat{\mathrm{D}}|\mathrm{I})$. Then the training maximizes the log-likelihood of counting while minimizing the KL divergence:

$$\hat{\Theta} = \arg\max_\Theta \mathbb{E}_{q_\Theta(\hat{\mathrm{D}}|\mathrm{I})}[\log(p(\mathrm{c}|\hat{\mathrm{D}}))] - \beta D_{KL}(q_\Theta(\hat{\mathrm{D}}|\mathrm{I})|p(\mathrm{D})) \tag{12}$$

Equation 12 has the same form as the $\beta$-VAE (Higgins et al., 2017) with four major differences. Firstly, this formula computes the count of objects instead of regenerating the input image. Secondly, no random variables are involved during training (equation 8), so we do not need to use the

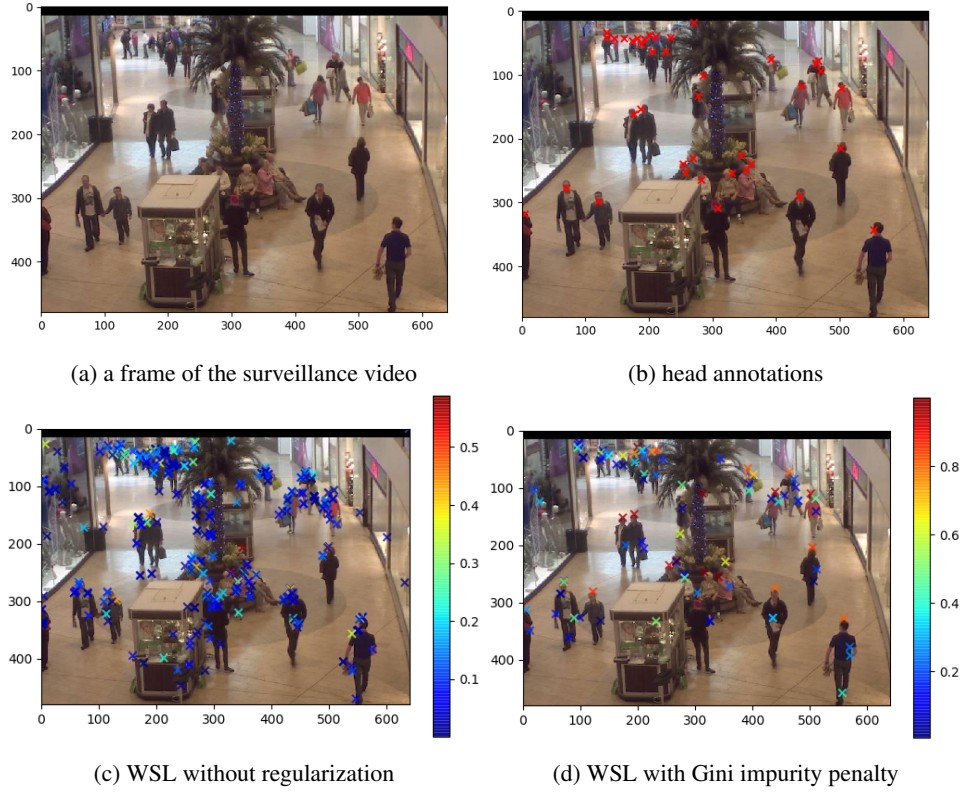

(a) a frame of the surveillance video

(b) head annotations

(c) WSL without regularization

(d) WSL with Gini impurity penalty

Figure 6: Mall data weakly supervised localization
(images are publicly accessible for research purposes)

reparameterization trick (Alemi et al., 2016). Thirdly, the latent variable of a VAE is usually a 1-D vector in contrast to a 2-D density map in this problem. Finally, the prior in a VAE is often set to be an isotropic Gaussian distribution (Burgess et al., 2018), whereas we do not know the exact distribution of the true density map; we only know its Gini impurity.

# 4  MALL DATA WSL

Similar to the variational term of the $\beta$-VAE, the Gini impurity penalty degrades the reconstruction (counting) performance, but it also encourages the density map to extract interpretable features. Here we take the Mall dataset as an example. The Mall set is a 2000-sample video clip collected through a surveillance camera above the hall of a shopping mall (Figure 6a). Although the heads of people are annotated (Figure 6b) by hand for counting and localization Loy et al. (2013); Chen et al. (2013); Change Loy et al. (2013); Chen et al. (2012), we aim to find the them from only the count (WSL).

Because human figures and the environment of the shopping mall are more complicated, we build up another FCNN for this task. It is a concatenation of 7 Residual Blocks (Res-block) He et al. (2016) as shown in Figure 7. The first convolutional layer of the Res-block increases or decreases the depth of its input tensor. Whereas the second layer maintains its inputs shape, so that the outputs of these two layers can be added together. To accelerate training, we apply batch-normalization Ioffe & Szegedy (2015) to the output of each Res-block. As before, no pooling is used here. Details of this FCNN are shown in Table 2. We use the first 1900 samples of the Mall set for training and the last 100 for validation. The original resolution of the Mall data is $480 \times 640$, but at the training stage we randomly crop a $300 \times 300$ patch of it to reduce computations. This is viable because a trained FCNN can be generalized to inputs of arbitrary sizes.

Table 2: FCNN for crowd counting

| operation | kernel | size | channels |
|---|---|---|---|
| Res-block | 3×3 | 300×300 | 32 |
| Res-block | 3×3 | 300×300 | 64 |
| Res-block | 3×3 | 300×300 | 128 |
| Res-block | 3×3 | 300×300 | 128 |
| Res-block | 3×3 | 300×300 | 128 |
| Res-block | 3×3 | 300×300 | 64 |
| Res-block | 3×3 | 300×300 | 32 |
| conv ($f(I)$) | 3×3 | 16×16 | 1 |

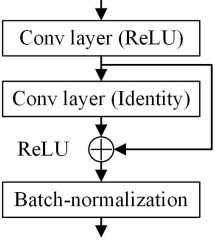

Figure 7: Residual Block

We trained the network according to equation 2, where the ground-truth count within a 300×300 patch is computed by adding up the head annotations. The counting error converges to around 0.85 after $3 \times 10^5$ epochs. An example of the hidden density map is shown in Figure 6c. It shows that the CNN pays more attention to the human figures than the irrelevant backgrounds. But the network does make some mistakes by counting parts of the booth, the tree, reflections, and the models as human beings, and for most of the bodies the detected features are along the their left sides. It can also be noticed that for an individual person, the highlighted points concentrate on the upper body. This could be because we used the appearances of heads for counting.

However, it is not clear by what discriminative features the network detects a human figure. Therefore, we decrease the Gini impurity ($\beta$=10.0) to reduce the amount of dots in the density map. After $3 \times 10^5$ epochs, the counting error converges to around 1.4, which is higher than before because of the regularization. The hidden density map is shown in Figure 6d. Compared with the previous result, the detected features concentrate over the heads of people. Herein, we can conclude that the regularized network counts human figures by detecting their heads. However, it is also noticed that the network struggles at detecting those who are remote or sitting on the bench. We believe this is because crowds standing far away have lower resolutions, and those on the bench sat still for a long time. Besides, they both have more overlaps than walking individuals.

Compared to class-guided weakly supervised localization, our method shows how specific features of a certain type of object are used for detecting. For human detection, the features could be the heads, hands, feet, etc., but some of them may not be used for the final counting. With the Gini impurity penalty, we are able to manipulate these features for better interpretation.

## 5 CONCLUSION

Because a deep CNN can maintain spatial information throughout very deep layers, a CNN trained with image-level labels has the potential of localizing detected targets. This paper uses a density-map-based counting machine to study this weakly supervised localization problem. First of all, the spatial information loss caused by pooling operations has been shown. This does not mean pooling cannot be used in any WSL networks, but for a plain end-to-end structure the lost information can hardly be recovered. Secondly, a simple technique has been applied to enhance the models robustness to rotation. By doing this, the detected features are concentrated within the object instead of along certain edges. Thirdly, a Gini impurity penalty has been introduced to control the amount of discriminative features in the hidden density map. This regularization acts in a similar way to the variational term of a $\beta$-VAE. Unlike a traditional $\beta$-VAE, which controls the capacity of a 1-D hidden space, this Gini impurity penalty restricts the capacity of a 2-D layer. Its potential in unsupervised learnings could be further explored in the future.

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
