# OpenReview forum: "Count-guided Weakly Supervised Localization Based on Density Map"
_ICLR.cc/2020/Conference — Reject_

### Official Review · AnonReviewer1 · 2019-10-08
**Official Blind Review #1**

**Rating:** 1

**Review:**

This article proposes a method for object counting which can be trained with weak supervision. Object counting methods are often trained with point annotations, i.e., one click-point per object. In this article, a weaker way of annotation is used: count-based annotation, i.e., the number of objects of each class present in the image are given as annotation but no precise location of the objects. This article is an extension of density-based object counting methods for weakly supervision.

This article analyzes the effect of pooling layers on WSL. The findings indicate that the pooling layer constrained the objects' locations to a predefined grid and accordingly they remove it. When removing this effect is alleviated. Moreover, the authors realized that a sift on the predictions also happen. Accordingly, they learn rotated versions of the model for centering the predictions. As a countereffect, several points per object are detected, and then a gini impurity regularization is proposed to reduce the number of detected objects. Then, the authors connect this formulation with VAE.

The experiments are conducted on a toy example, i.e., circle finding, and on the well established Mall dataset.

Main concerns:
The experimental section should make experiments on standard datasets (i.e., USCD, Trancos, Mall, PkLot, Shangai, Penguins) using standard evaluation protocols (MAE, GAME). The current evaluation only shows the qualitative results on one frame of the Mall dataset.

The results of the proposed method should be compared with baselines. For instance, Glance method, which is only trained with count-level annotation, and with "C-WSL: Count-guided Weakly Supervised Localization". It should be also compared with WSL methods based on image-level labels such as "Object Counting and Instance Segmentation with Image-level Supervision" or "Where are the Masks: Instance Segmentation with Image-level Supervision" and then with methods that use point supervision as an upper bound such as density method or Where are the blobs.

Moreover, some ablation studies about the effect of each component of the model would be needed.

Regarding the model I have some concerns:
 - Removing the pooling layers does not seem a very good idea. This will remove part of the ability to detect objects at multiple scales and moreover will increase its computational power. Maybe for easy cases as circles and heads of mall people where the resolution changes are small could work. But not for more challenging datasets.
 - The rotation of the model could be ok for circles or heads. But making the model to learn rotation-invariant features could be challenging in more difficult datasets.
 - Eq.5 and 8 which are used in the model seem to use point supervision and not just the count based. Which makes the method not a weakly supervised one. Are these equations the ones used or others?



**Experience Assessment:**

I have published in this field for several years.

**Review Assessment: Checking Correctness Of Derivations And Theory:**

I did not assess the derivations or theory.

**Review Assessment: Checking Correctness Of Experiments:**

I carefully checked the experiments.

**Review Assessment: Thoroughness In Paper Reading:**

I read the paper at least twice and used my best judgement in assessing the paper.

---

### Official Review · AnonReviewer2 · 2019-10-23
**Official Blind Review #2**

**Rating:** 3

**Review:**

This paper presents a method to train a network for counting/localizing objects in a weakly supervised framework.
The network is optimized based on weak supervision about target object counts; training images have only the ground truth of the total number of object counts without their positions in images.
Through analyzing the feature maps of the network trained on toy bubble images, the authors propose two regularization techniques for rotation robustness and sparseness of the map in order to improve  performance of object localization in the feature map.
The experimental results on object counting tasks using the Mall dataset show that the proposed method produces favorable performance.

This paper is leaning toward rejection due to the following two reasons.
(1) The presented techniques are limited to the specific task of object counting.
(2) They are derived in an ad-hoc way based on less theoretical background and thus lack novelty.

The detailed comments are as follows.

* The technique to embed rotation invariance by Eq.(4) is presented in an ad-hoc way. This seems to be limited to the tasks on toy data such as bubble images shown in the paper. In the real-world tasks/images to count the more complicated objects, the spatial correlation among object parts is an important clue to provide discriminative features for detecting objects such as torso below the head in human figures. Such spatial dependency could be missed by imposing the rotation invariance on the network.

* The regularization by Gini impurity is a well-known technique to induce sparsity, lacking novelty. There is a large body of researches for enhancing sparsity, and thus to validate the regularization, the authors should compare the Gini impurity with the other sparsity-inducing regularization such as entropy. For example, it could be possible to argue the regularizations in terms of their derivatives (gradients) to be used for back-propagation. This paper lacks detailed analysis and discussion about the regularization in this counting framework.

* It is unclear how the perspective from the (beta-)VAE contributes to the analysis of the proposed method. There seems to be less connection between Eq.(8) and the object counting task/framework. The authors just show the similarity between the presented method and VAE in terms of formulation. Since the proposed method belongs to just a simple optimization with sparsity-inducing regularization, the reviewer cannot find any convincing reason to discuss the connection to VAE.

* The method is not fully validated in the experiments. The authors provide only one experimental result on the Mall dataset which is insufficient to validate the effectiveness of the proposed method. Considering that the method is limited to the specific task of object counting, it is necessary to qualitatively and thoroughly evaluate the performance on various datasets that exhibit various environmental conditions regarding such as lighting and occlusion.

Minor comments:
- Show the index for summation in Eqs.(1,4,6,7,8).
- It is confusing to show q_\Theta(D|I) in Eq.(3), even though the authors aim to discuss the connection to VAE. Show the mathematical definition of q_\Theta(D|I) before that.

**Experience Assessment:**

I have read many papers in this area.

**Review Assessment: Checking Correctness Of Derivations And Theory:**

I carefully checked the derivations and theory.

**Review Assessment: Checking Correctness Of Experiments:**

I carefully checked the experiments.

**Review Assessment: Thoroughness In Paper Reading:**

I read the paper thoroughly.

---

### Official Review · AnonReviewer3 · 2019-10-29
**Official Blind Review #3**

**Rating:** 3

**Review:**

The main contribution of the paper is the extension of techniques for weakly supervised localization, i.e. given ground truth counts of objects in a given image, one can do training to generate hidden layer density maps that allow for feature detection and localization of objects.   The main contribution of the paper seems to be the regularization of the density map by incorporation of a Gini impurity penalty and the contrasting of the regularizer against beta-variational autoencoder formulation. The experiments show the utility of the method in a toy example and an example involving pedestrians in video surveillance.

While the extensions are interesting, I find that the paper contributions are incremental and do not offer sufficient experimental results. The experiments are qualitative (just illustrations of results on a few images).  While I understand the context of the contribution from authors point of view and can appreciate the study of weakly-supervised learning, I find it important to ask how the present framework compares with a traditional method that uses background subtraction (thus gaining appearance invariance to an extent) followed by geometric prior driven indexing of counts and locations (see for example:  Lan Dong et al (ICCV 2007 - Fast Crowd Segmentation via Shape Indexing). The inductive bias and interpretability of design is extremely transparent and obvious.  Can you please elaborate how your work compares in performance to such a technique and what you mean by feature interpretability in your context?

My rating is based on the fact that the results are preliminary and further quantitative experimentation is needed.

**Experience Assessment:**

I have read many papers in this area.

**Review Assessment: Checking Correctness Of Derivations And Theory:**

I did not assess the derivations or theory.

**Review Assessment: Checking Correctness Of Experiments:**

I assessed the sensibility of the experiments.

**Review Assessment: Thoroughness In Paper Reading:**

I read the paper at least twice and used my best judgement in assessing the paper.

---

### Decision · Program_Chairs · 2019-12-19

**Decision:**

Reject

**Comment:**

This work proposes a new regularization method for weakly supervised localization based on counting.
Reviewers agree that this is an interesting topic but the experimental validation is weak (qualitative, lack of baselines), and the contribution too incremental.
Therefore, we recommend rejection.